# The structural basis of translational control by eIF2 phosphorylation

Tomas Adomavicius[1], Margherita Guaita[1], Yu Zhou[1,2], Martin D. Jennings [1], Zakia Latif[1,3], Alan M. Roseman[1] & Graham D. Pavitt[1]

Protein synthesis in eukaryotes is controlled by signals and stresses via a common pathway, called the integrated stress response (ISR). Phosphorylation of the translation initiation factor eIF2 alpha at a conserved serine residue mediates translational control at the ISR core. To provide insight into the mechanism of translational control we have determined the structures of eIF2 both in phosphorylated and unphosphorylated forms bound with its nucleotide exchange factor eIF2B by electron cryomicroscopy. The structures reveal that eIF2 undergoes large rearrangements to promote binding of eIF2α to the regulatory core of eIF2B comprised of the eIF2B alpha, beta and delta subunits. Only minor differences are observed between eIF2 and eIF2αP binding to eIF2B, suggesting that the higher affinity of eIF2αP for eIF2B drives translational control. We present a model for controlled nucleotide exchange and initiator tRNA binding to the eIF2/eIF2B complex.

[1] Division of Molecular and Cellular Function, School of Biological Sciences, Faculty of Biology, Medicine and Health, Manchester Academic Health Science Centre, The University of Manchester, Manchester M13 9PT, UK. [2] Present address: The Jenner Institute, Old Road Campus Research Building Roosevelt Drive, Oxford OX3 7DQ, UK. [3] Present address: Department of Microbiology and Molecular Genetics, University of the Punjab, Lahore, Pakistan. Correspondence and requests for materials should be addressed to A.M.R. (email: alan.roseman@manchester.ac.uk) or to G.D.P. (email: graham.pavitt@manchester.ac.uk)

Eukaryotic protein synthesis typically begins with a specialised initiator methionyl transfer RNA (Met-tRNAi) that is delivered to ribosomes by the translation factor eIF2 as part of a larger preinitiation complex (PIC) with multiple other translation initiation factors[1]. Within the PIC eIF2 also helps ensure that start codons are accurately recognised. The affinity of Met-tRNAi for eIF2 is controlled by guanine nucleotides. They interact with high affinity only when eIF2 is bound to GTP[2,3]. GTP hydrolysis is triggered by the GTPase-activating protein eIF5 and Pi release is prompted by AUG codon recognition within the PIC, forming an eIF2-GDP complex with low affinity for Met-tRNAi. Hence, eIF2-GDP leaves the ribosome together with eIF5[4,5], and here eIF5 inhibits spontaneous GDP release[6]. Only by re-engaging with GTP can eIF2 participate in further rounds of Met-tRNAi binding and protein synthesis initiation[1]. This requires the action of eIF2B. eIF2B first removes eIF5[7] and then acts as a guanine nucleotide exchange factor (GEF) to activate eIF2 and facilitate Met-tRNAi interaction and rebinding of eIF5[3]. This last step prevents eIF2B competing and destabilising eIF2-GTP/Met-tRNAi ternary complexes (TCs)[3,8]. Thus, eIF2 activation is critically important for translation initiation.

eIF2 activation is highly regulated. In response to a wide range of signals, multiple protein kinases phosphorylate a single serine, historically known as Ser51, within the eIF2α subunit. This inhibits the GEF activity of eIF2B forming a tight eIF2αP/eIF2B inhibitory complex[3,9,10]. As eIF2B levels are lower than eIF2 in cells, partial phosphorylation is sufficient to attenuate protein synthesis initiation[1]. A range of stress-responsive messenger RNAs (mRNAs) are resistant to, or stimulated by, reduced TC levels[11,12]. The response is generally termed the integrated stress response (ISR)[13,14]. It is now clear that aberrant ISR responses are intimately linked to a wide range of human diseases[15] and are a potential therapeutic target[14].

Structural biology approaches have recently made important contributions to our understanding of many steps of protein synthesis initiation, including how TC interacts with other factors and the small ribosomal subunit[16]. Structural studies of eIF2B have shown that it is a decamer or a dimer of pentamers[17–19]. eIF2B has a central hexameric core comprising an eIF2Bα homodimer and (βδ)₂ heterotetramer that is linked to a pair of γε heterodimeric arms. Prior genetic and biochemical evidence implicates the central core as critically important regulatory subcomplex (RSC) for sensing eIF2α Ser51 phosphorylation by direct eIF2α binding[10,20]. In contrast, the eIF2B GEF domain is found at the eIF2Bε carboxyl terminus[21,22]. This domain is sufficient for minimal GEF action in vitro[21,22], and its activity is stimulated by interactions with the other eIF2B subunits, principally eIF2Bγ for the yeast factor, although human eIF2B may require all subunits for full activity[7,18,20,23].

Although structures of eIF2 and eIF2B have been determined[17,24], the structural basis of GEF action and how it is controlled by eIF2 phosphorylation remain unclear. Here we have used single-particle electron cryomicroscopy (cryoEM) to resolve the structures of eIF2αP/eIF2B and eIF2/eIF2B complexes from Saccharomyces cerevisiae to an average resolution of 3.9 and 4.6 Å, respectively. We show that the eIF2B decamer binds to two molecules of eIF2αP simultaneously, one at each side. eIF2α undergoes extensive conformational change from its TC form to dock with eIF2B, which exhibits only minor changes in structure compared with free eIF2B. The phosphorylated eIF2α subunit makes extensive contact with a regulatory interface dominated by eIF2Bα and eIF2Bδ, which agrees with previous genetic and biochemical observations. Our structural analysis provides a molecular explanation for how these two factors interact and how eIF2 phosphorylation locally modifies the eIF2α regulatory loop that contributes to local differences between eIF2 and eIF2αP

binding to eIF2B. We provide a model for how changes in eIF2 and eIF2B interactions may promote both GEF action and facilitate coupled recruitment of initiator tRNA to eIF2-GTP. Finally, a combination of structural similarities and differences between eIF2α's interactions with eIF2B and the double-stranded RNA-activated protein kinase (PKR) are observed. These findings help explain why the Vaccinia protein K3L, a structural mimic of eIF2α, acts as a pseudo-substrate inhibitor of PKR without also inhibiting eIF2B. This work provides molecular insight into a cellular regulatory mechanism that is central to the ISR.

## Results

**Structure of phosphorylated eIF2 in complex with eIF2B.** We made use of our previously described expression and purification schemes that use yeast cells to separately purify active S. cerevisiae eIF2B and eIF2 protein complexes free from each other (Supplementary Fig. 1a)[3,25]. The yeast strains used are deleted for the sole eIF2α kinase Gcn2; hence, eIF2 is purified uniformly dephosphorylated at the ISR regulatory site. As phosphorylated eIF2 has ten-fold higher affinity for eIF2B than unphosphorylated eIF2 ($K_d$, 3.5 vs. 32.2 nM)[3], we first focused on this complex. Purified PKR kinase was used to stoichiometrically phosphorylate eIF2α in vitro (Supplementary Fig. 1b). eIF2αP/eIF2B complexes were generated by mixing the purified proteins and fractionating them by size exclusion chromatography. The resulting complex size (~1 MDa) is indicative of a 2:1 eIF2/eIF2B complex (Supplementary Fig. 1c).

Protein samples were vitrified on grids and images recorded by cryoEM (Supplementary Table 1). Our initial attempts at three-dimensional (3D) classification and reconstruction revealed an orientation bias in the sample that precluded generation of a 3D model. To solve this issue, we changed grid type and collected images using a 35° tilted stage. When combined, our data successfully resulted in the range of images required for 3D reconstruction (Supplementary Fig. 2a). The central core of the structure exhibited clear two-fold rotational symmetry in two-dimensional (2D) projection classes (Supplementary Fig. 2b). An initial three-dimensional (3D) map had defined density at the centre, but was more diffuse laterally (Supplementary Fig. 2c). We therefore refined the core of the structure applying a mask to exclude the variable peripheral features and generated a 3.9 Å map, into which a homology model of the S. cerevisiae eIF2B decamer based on the S. pombe crystal structure (PDB 5B04) could be docked (Supplementary Fig. 2d)[17]. Extensive local adjustments were made to the model, guided by the density. Example density fitting is shown in Supplementary Fig. 2e. The amino-terminal domains 1 and 2 (NTD) of S. cerevisiae eIF2α[26] were fitted within the remaining density, with local refinements (Supplementary Table 2). Saccharomyces cerevisiae eIF2γ and the eIF2α carboxy-terminal domain could be docked as rigid bodies, at lower resolution, into the diffuse density at the sides of the high-resolution centre[24]. Fourier shell correlation (FSC) analyses demonstrate good correlation between the cryoEM map and the atomic model and the absence of overfitting (Supplementary Fig. 2f). Modelling statistics are shown in Supplementary Table 3.

The final model shows two eIF2αγ complexes with a minimal eIF2β NTD helix, each bound at one side of a central eIF2B decamer (Fig. 1). The resolution varies from 3.5 Å at the core to 18 Å at the periphery (Supplementary Fig. 2g). Each eIF2αP NTD makes extensive contact with the hexameric regulatory core of eIF2B comprised of an eIF2Bα dimer and eIF2Bβδ heterodimers. Each eIF2αP is inserted between one eIF2Bα and 2Bδ and also makes contact with the adjacent eIF2Bβ to anchor eIF2αP to eIF2B (Fig. 1). In contrast, each eIF2γ makes looser or transient contact with one adjacent lateral eIF2Bγε arm (see below). eIF2γ

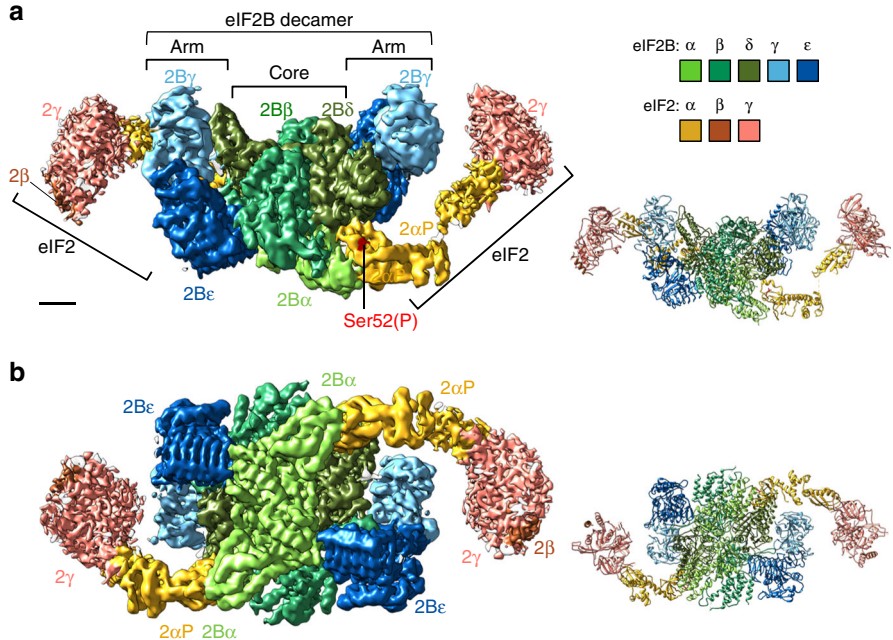

**Fig. 1** Electron cryomicroscopy (cryoEM) structure of the eIF2B/eIF2αP complex. Left: Refined 3.9 Å cryoEM map of eIF2B/eIF2αP complex with central eIF2B decamer and two lateral eIF2 trimers. Local amplitude scaling (Locscale) was used to apply local density re-scaling based on a fitted model. Right: modelled densities displayed as secondary structures. **a** Top and **b** back views shown. Map surface and subunits are coloured as indicated, with the regulatory phosphoserine in red. Scale bar relates to cryoEM map and is 20 Å. The figure was drawn with UCSF Chimera software

is the GDP/GTP binding subunit and is orientated so that the nucleotide-binding surface faces inwards towards eIF2B. This mode of binding is consistent with prior genetics and biochemistry[10,20,27,28], including many cross-links identified in recent experiments[17], but is distinct from recently published models predicting how eIF2 and eIF2B may interact[17,29,30]. As in cells eIF2B levels are limiting, the 1:2 eIF2B:eIF2 structure is fully consistent with partial eIF2 phosphorylation in vivo being able to fully repress eIF2B function and protein synthesis initiation.

**Conformational changes in eIF2αP and in eIF2B on binding.** The observed conformation of eIF2 in the eIF2αP/eIF2B complex is markedly different from structures of the Met-tRNAi-bound TC as found within the yeast PIC[24] (Fig. 2a). To adopt this position, an elbow-like rotation between eIF2α domains 2 and 3 must occur between these distinct ligand-bound states. Such large-scale movement is consistent with observations of eIF2α flexibility between domains 2 and 3, seen in solution nuclear magnetic resonance experiments of isolated human eIF2α[30,31]. Hence, eIF2α domain flexibility appears biologically important for distinct eIF2 ligand interactions. Comparison of eIF2αP domains 1 and 2 with prior eIF2α structures reveals high agreement between other yeast, rabbit and human structures (root mean square deviation (RMSD) <2.5 Å; Supplementary Table 4). However, upon eIF2B binding there is a local change. eIF2α residues 58–64 form a clear two-turn α-helix that is not observed in structures lacking eIF2B (Supplementary Fig. 3a). This helix forms a stable structure that places the main chain in a position so that residues here interact directly with eIF2Bδ (Fig. 3a, b; see next section).

Unlike the tight binding of eIF2αP to the regulatory eIF2B core, eIF2γ interacts more transiently with an eIF2Bγε arm. It is observed in multiple positions due to eIF2α flexibility. This flexibility is captured in a series of lower-resolution maps that each trap distinct conformations of eIF2αP (Fig. 2b–g) that are all different to the eIF2α conformation in the TC (Fig. 2h)[24]. These maps were produced using a localised reconstruction script (see Methods) to isolate and superpose the two independent halves of each image. Classification was performed with no image alignment yielding ten 3D classes with relatively even particle distributions (Supplementary Fig. 3b). The classes shown in Fig. 2 represent the full range of eIF2α conformations identified. *Saccharomyces cerevisiae* eIF2γ/α domain 3 structures were docked into each class map. The maps reveal a multitude of eIF2αP (domain 3)-eIF2γ conformations relative to the core eIF2B/eIF2αP (domains 1 and 2), possibly suggesting a continuous flexing of the eIF2αP (domain 3)-eIF2γ arm (Fig. 2b–g). When linked as movie frames, they indicate that eIF2αP provides a dominant stable interface to the regulatory eIF2B core while interaction with the eIF2B catalytic arm appears transient (Supplementary Movie 1). eIF2α domain 3 undergoes a 46° rotation between the extreme states (Supplementary Fig. 3c). These observations are entirely consistent with the idea that tight binding to eIF2αP limits both release of eIF2 from eIF2B and productive interaction of the eIF2Bε GEF domain to limit/impair overall GEF activity[22]. The stable eIF2α interactions and weaker variable eIF2γ binding likely contribute significantly to the mechanism of translational control.

In contrast to the large domain rearrangements observed in eIF2α, the eIF2B decamer appears to have relatively modest changes when our structure is compared with previous eIF2B decamer structures. The subunits in our structure are highly similar to both the *S. pombe* crystal structure[17] and human eIF2B cryoEM structures where the compound ISRIB is bound to the eIF2Bββ core[18,19]. When superposed, each subunit differs by RMSD <1.7 Å, except for the eIF2Bγ subunits (2.5–2.7 Å) where resolution is poorer (Supplementary Table 5). Comparing the structures globally, we observe that small changes to the core subunit orientations appear to propagate through to the catalytic arms that may be attributable to eIF2αP binding. When our eIF2-bound structure was compared to the unbound *S. pombe* decamer, each eIF2Bεγ arm appears to open up by up to 7° along the front axis and additionally rotate by 7° along the view from one arm (Supplementary Fig. 3d). These observations suggest that structural rearrangements upon eIF2α binding at the

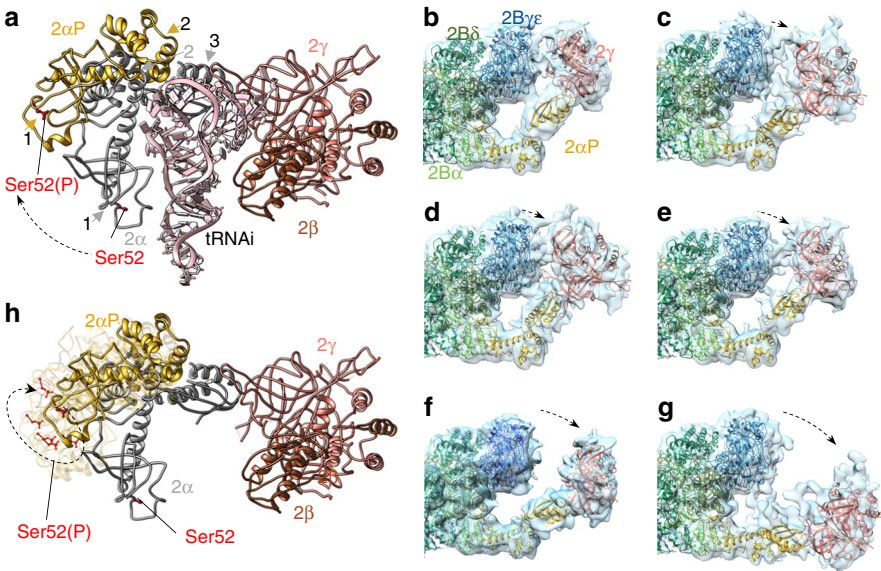

**Fig. 2** eIF2α conformational flexibility on binding to eIF2B. **a** eIF2α conformation differs between eIF2αP/eIF2B and ternary complexes (TCs) complexes. eIF2α from our eIF2αP/eIF2B complex (domains 1 and 2 shown in gold and arrowed, domain 3 grey) aligned onto TC from 3JAP (2α domains 1–3 in grey) using eIF2α domain 3 as a reference. **b–g** Flexibility between eIF2α domains 2 and 3 seen in eIF2αP/eIF2B 3D classes obtained when halves of the particles were independently classified using a localised reconstruction script (described in the Methods section and Supplementary Fig. 2b). **h** eIF2αP conformations from **b–g** aligned modelled onto 3JAP eIF2 (as in **a**) as semi-transparent ribbons. Dashed black arrows indicate changed positions. In **a**, **h**, Ser52 and Ser52(P) side chains are shown in red

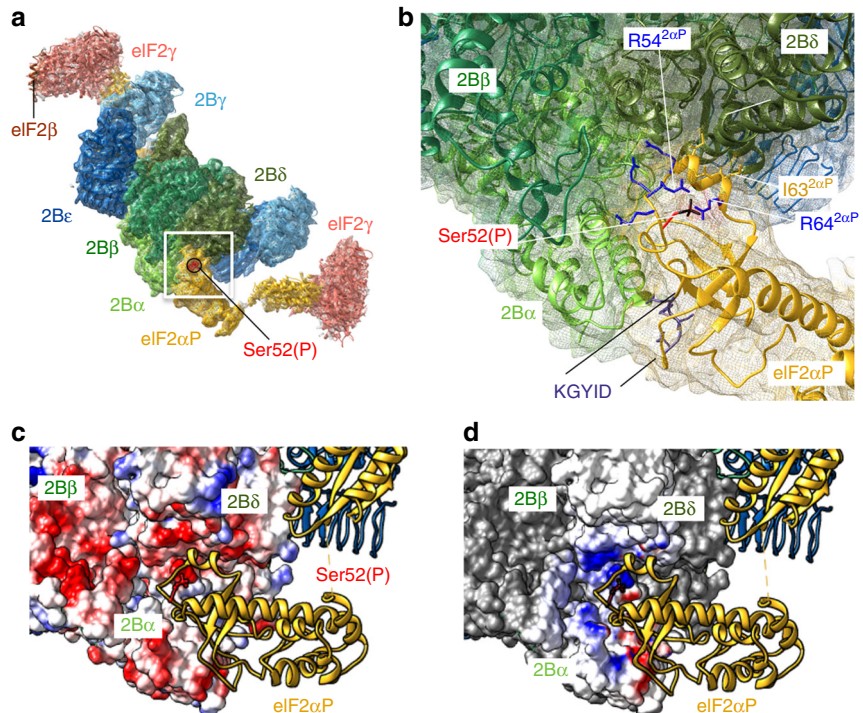

**Fig. 3** eIF2α N-terminal domain binds between the eIF2B α and δ subunits. **a** Overview and **b** detail of eIF2αP molecular interactions with eIF2B, showing the electron cryomicroscopy (cryoEM) density as a mesh and highlighting a network of arginines around Ser52 and the KGYID sequence. **c**, **d** Complimentary surface potentials at the eIF2αP/eIF2B interface. **c** Electrostatic surface representation of eIF2Bαβδ and **d** Coulombic potential due to eIF2αP displayed at the eIF2Bαβδ surface. Blue, positive and red, negative

regulatory core may be transmitted through the whole eIF2B decamer to potentially influence GEF activity. However, at this time, we cannot rule out the possibility that the eIF2B changes are attributable to species variation or cryoEM vs. crystallographic method constraints.

**The eIF2αP/eIF2B phospho-regulatory core interface.** The regulatory phosphoserine (serine 52 in *S. cerevisiae* and *Homo sapiens* eIF2α) sits in a conserved loop within domain 1 (residues 48–57) that contributes to the interface with eIF2B and is well resolved. Ser52 itself does not contribute directly to the interface

with eIF2B, and the phosphate remains surface exposed. The positively charged side chains R54 and R64 are angled towards Ser52(P) and likely help stabilise this conformation of this important loop of eIF2α. All eIF2α residues in contact with the eIF2B core are conserved between the yeast and human proteins. The eIF2α loop containing Ser52(P) makes contact with three eIF2B subunits: eIF2Bα, eIF2Bδ and a minor eIF2Bβ contact (Fig. 3b, Supplementary Fig. 4c).

Genetic and biochemical experiments have identified a large series of missense mutations in eIF2α that compromise translational control by eIF2αP in what is known as general amino-acid control, the yeast analogue of the mammalian ISR[32,33]. Termed Gcn⁻ mutations, missense alleles in eIF2α have been classified as affecting the ability of eIF2 kinases, including Gcn2 and PKR, to phosphorylate eIF2 and/or to impair eIF2B interactions[34,35]. Among these are conserved eIF2α residues 80–84 (sequence KGYID) that form an important interface between the eIF2 kinases and eIF2α as demonstrated by studies examining the genetic and biochemical impact of mutations and the co-crystal structure of PKR and eIF2α[36,37]. Our structure now reveals that there is extensive overlap between the eIF2Bα and PKR interfaces with eIF2α (Supplementary Fig. 4), such that each interaction is likely to be mutually exclusive (see below).

Gcn⁻ missense mutants were also previously identified within the yeast eIF2Bα, β and δ subunits[27,38]. Many affect residues located at the interfaces between eIF2B subunits themselves, as indicated previously[17]. However, in eIF2Bα side chains of T41 and E44 contact eIF2α D84 and Y82, respectively (non-H atoms are within 4 Å). Both eIF2α residues are within the important KGYID element. Mutation of any of these four residues confers a Gcn⁻ phenotype consistent with the importance of this contact site for phospho-regulation of eIF2B activity (Supplementary Fig. 5c)[27,35]. Other Gcn⁻ mutants in eIF2Bδ (E377K and L381Q) disrupt both yeast and mammalian eIF2 phospho-regulation despite allowing efficient phosphorylation of Ser52[9,27]. Here, δE377 and δL381 are seen to contact eIF2α I59 and I63, respectively. This, along with L62, represents an eIF2B/eIF2α-specific interface, that is, not shared with PKR (Supplementary Figs. 4 and 5). Support for the importance of this eIF2Bδ/eIF2α contact comes from a previously unpublished genetic suppressor analysis. A novel missense mutation eIF2α[I63N] was isolated, which specifically suppresses the Gcn⁻ phenotype of the eIF2Bδ[L381Q] mutant strain enabling robust growth of eIF2α[I63N] eIF2Bδ[L381Q] double mutant cells following amino-acid starvation (Supplementary Fig. 5a, row 4). The I63N mutation does not suppress the amino-acid starvation induced growth sensitivity observed with other eIF2Bδ or 2Bα mutants tested (Supplementary Fig. 5a), or impair the ability of the kinase Gcn2 to phosphorylate eIF2α (Supplementary Fig. 5b). This demonstrates an allele-specific suppression of the eIF2Bδ[L381Q] mutant phenotype by the eIF2α-I63N mutation (Supplementary Fig. 5c). Although eIF2Bβ is also in contact with the Ser52 loop and Gcn⁻ alleles affect this subunit, our eIF2Bβ density is weaker in this region and residues mutated previously do not make direct contact with eIF2α.

To further test the idea that PKR and eIF2B compete for the same binding site on the surface of eIF2α, we asked whether eIF2B could compete with PKR for access to eIF2α in an in vitro kinase assay. Phos-tag acrylamide gels separate eIF2α into phosphorylated and unphosphorylated forms according to the extent of eIF2α phosphorylation. We find that eIF2B can antagonise the ability of PKR to phosphorylate eIF2α within purified eIF2 in a concentration-dependent manner (Fig. 4). At low eIF2B concentrations, PKR can phosphorylate eIF2α well (Fig. 4a, lanes 4–8 and Fig. 4b), but as eIF2B and eIF2 concentrations approach the 1:2 eIF2B:eIF2 stoichiometry

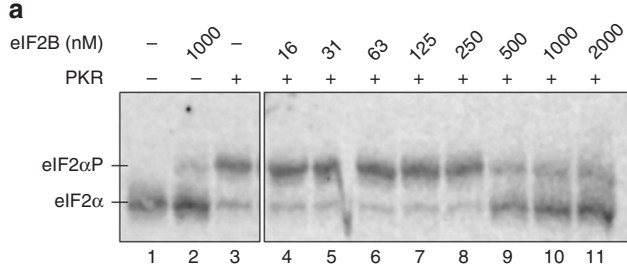

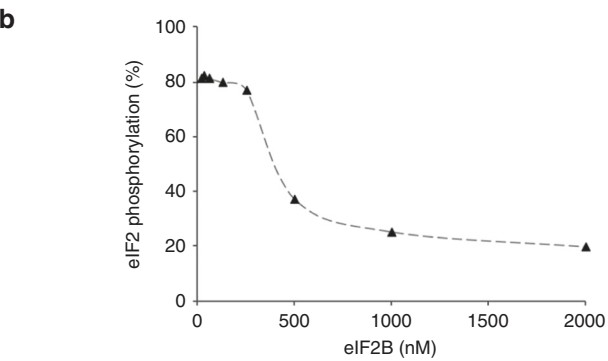

**Fig. 4** eIF2B antagonises RNA -activated protein kinase (PKR) activity. **a** Western blot of eIF2α phosphorylation (1 μM eIF2) by PKR (10 nM), when incubated in the presence of increasing concentrations of eIF2B. eIF2α and eIF2αP resolved by Phos-tag acrylamide gel electrophoresis. Both panels are from the same blot. Experiment repeated four times with similar results. Source data are provided in the Source Data file. **b** Quantification of eIF2αP percentage

observed in our structure (500 nM eIF2B), or eIF2B is in excess (lanes 9–11), PKR action is antagonised. These data support the structural and earlier genetic/biochemical findings and are fully consistent with the conclusion that eIF2B and PKR compete for an overlapping binding interface on eIF2α.

**Shared eIF2αP/eIF2B and eIF2 PIC interfaces.** Structures of the partial yeast PIC have revealed interactions between eIF2α, Met-tRNAi, mRNA and the ribosomal proteins uS1, uS7 and uS11[24]. Our comparative analysis indicates that many surface residues of eIF2α contribute to interactions between eIF2α and the core eIF2B subunits, as well as to PIC components (Supplementary Fig. 4). Specifically, the eIF2α KGYID sequence makes contacts with uS11 within the PIC as well as the previously noted eIF2Bα and PKR interactions. Similarly, Met-tRNAi binding surface within the 48S PIC structure partially overlaps with the eIF2Bδ binding surface. As overlapping surfaces of eIF2α contribute to interactions with multiple partners required for protein synthesis, this will place constraints on the range of regulatory alleles that can be identified in eIF2α by mutagenesis. For example, the arginine residues adjacent to S52 (R55 and R57) project into the junction formed between the three eIF2B subunits enabling eIF2Bα, β and δ to simultaneously contribute to eIF2αP recognition (Fig. 3b). Regulatory Gcn⁻ mutations were not identified here[35]. This may be because these residues also make important contributions towards eIF2α interactions within the PIC that preclude identifying alleles with a Gcn⁻ phenotype (Supplementary Fig. 4).

The eIF2α binding surface formed by eIF2Bαβδ is strongly negatively charged (Fig. 3c), while the interacting interface of eIF2αP is oppositely charged (Fig. 3d), suggesting that this provides a basis for strong binding and is in agreement with the salt sensitivity to their interaction[39]. Perhaps surprisingly, Ser52 (P) of eIF2α does not contribute directly to the eIF2B binding

interface. Instead, the Ser52(P) side chain remains surface exposed within the complex. Overall, our eIF2α/eIF2B structure provides insight into the molecular basis of the regulatory interface between these translation factors, one that is critical for the ISR. Our data are compatible with a model where eIF2 kinases and eIF2Bα compete for an overlapping interface on eIF2α, while eIF2Bδ extends the eIF2B interface, in line with previous genetics and biochemical findings[35].

**eIF2/eIF2B structure is almost identical to eIF2αP/eIF2B.** We used the approach described above to determine the structure of the non-phosphorylated eIF2/eIF2B complex to an overall resolution of 4.6 Å (Supplementary Fig. 6a–d, Supplementary Tables 1 and 3). The 3D map generated for this eIF2/eIF2B complex is remarkably similar to the eIF2αP/eIF2B complex, with two eIF2 molecules bound one at each side of the eIF2B decamer. Therefore, to build the atomic model for this complex, our eIF2αP/eIF2B core model (eIF2B with eIF2α domains 1 and 2) provided the initial eIF2α/eIF2B atomic coordinates. All local structural differences identified between this model and the eIF2/eIF2B core map were rebuilt and refined to generate the final eIF2/eIF2B core atomic model. The remaining less well-resolved lateral density was assigned to the remaining eIF2 subunits, which were rigid body fitted to provide our final model (Fig. 5a).

The overall similarity of the two eIF2/eIF2B complexes is clear from an overlay of the two cryoEM maps (Supplementary Fig. 6e, f). When our eIF2B decamer atomic models are aligned in their entirely, with each treated as a single molecule, the RMSD is 0.8 Å. Similarly, when equivalent individual eIF2B subunits are aligned optimally, they have an RMSD of only 0.6 Å (apart from eIF2Bγ which is 1.0 Å) overall Cα atoms matched (Supplementary

Table 5). Hence, the eIF2B subunit structural models are highly similar, with only very minor rearrangements.

When the Ser52 loop of eIF2α is examined closely, one clear local difference is loss of the density associated with the S52 phosphate. There is also some local rearrangement of the S52-containing regulatory loop in eIF2α (Fig. 5b). Specifically, R53 is reoriented in the unphosphorylated complex. In addition, there are minor movements associated with R55 and R64, the latter moves away from S52. The density around R54 is weaker in eIF2/eIF2B than in the eIF2αP/eIF2B complex structure, suggesting that it may adopt more than one position. Here we have shown R54 in its original position, but it may reorient away from this position as there is weak density in several compatible positions for this side chain. Overall, the weaker density in the S52 phospho-loop of the eIF2/eIF2B complex prevents us defining precisely the positions of the side chains. This likely points to enhanced flexibility of this region of eIF2α in the absence of the phosphate group. This interpretation is consistent with the observed ten-fold reduction in steady-state affinity between the proteins in the complex[3].

## Discussion
During the ISR, global protein synthesis initiation is repressed through the phosphorylation of translation initiation factor eIF2 and the formation of an inhibitory complex with eIF2B. Here we determined structures of the eIF2/eIF2B complex with and without eIF2 phosphorylation by cryoEM. We find that two molecules of the eIF2 heterotrimer bind laterally, one to each side of the eIF2B decamer. In both complexes we observe tight interactions between eIF2α and the eIF2Bαβδ regulatory core. eIF2α adopts a highly extended conformation in both complexes, distinct from its form when in complex with Met-tRNAi (Fig. 2). Phosphorylation of eIF2α causes only modest local rearrangements to the ser52 regulatory loop (Fig. 5b). In both complexes, we find a looser interaction between eIF2γ and the eIF2Bγε catalytic arms. We were not able to resolve the separate eIF2Bε GEF domain to high-resolution in either complex (see below).

Although highly surprising that there are only minor changes in conformations observed between the two complexes, these data do help explain recent observations. First, cross-linking experiments revealed no significant changes in sites of interaction between eIF2/eIF2B and eIF2αP/eIF2B from *S. pombe*[17] (see Supplementary Discussion). Second, we found previously that the affinity of eIF2 for eIF2B was not altered by the presence or absence of guanine nucleotides[3]. This finding was highly unexpected because GEFs typically favour interaction with the GDP or nucleotide-free forms of their G protein partners[40]. However, because nucleotides bind eIF2γ[41] and our structural data reveals that the dominant eIF2B interaction is with eIF2α, this helps explain eIF2B's apparent lack of nucleotide specificity in steady-state conditions, as we were likely measuring eIF2α/eIF2B binding stability rather than the eIF2γ/eIF2B interaction implicated in GDP release.

The eIF2Bε GEF domain was not resolved in both our high-resolution structures. However, within the lower-resolution half-particle reclassifications (Fig. 2b–g), we could identify a low-resolution 3D map class with additional density into which both additional eIF2β and the eIF2Bε GEF atomic models could be rigid-body fitted (Supplementary Fig. 7). Although the density is too weak for precise fitting, the 190 residue eIF2Bε GEF domain is composed of HEAT repeats and is at the carboxy terminus of eIF2Bε, joined via a likely highly flexible approximately 90 residue linker sequence[22]. eIF2Bε GEF can contact both eIF2β and eIF2γ[42–44]. Our docking model is consistent with these findings.

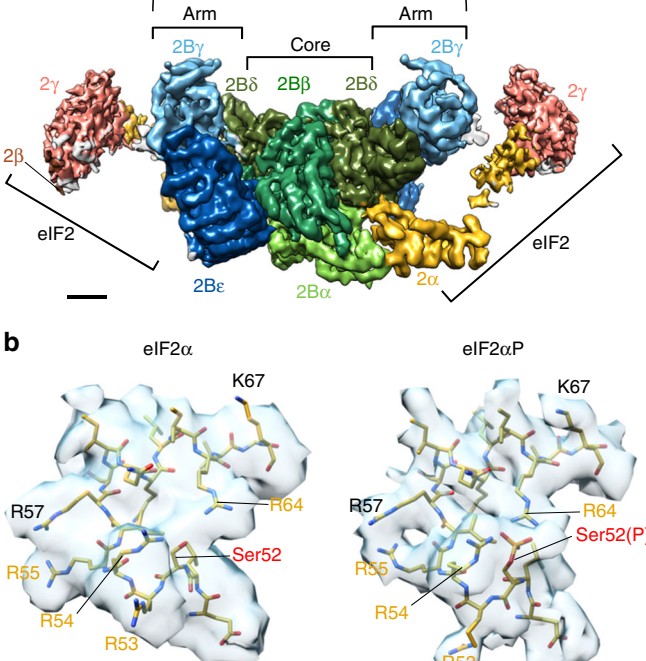

**Fig. 5** Non-phosphorylated eIF2/eIF2B complex is almost identical to eIF2αP/eIF2B. **a** Overview of eIF2/eIF2B structure map (after local amplitude scaling (Locscale)). Orientation and surface coloured as in Fig. 1a. Scale bar is 20 Å. **b** Model fitting to electron cryomicroscopy (cyoEM) maps around Ser52 in eIF2/eIF2B (left) and eIF2αP/eIF2B (right) complexes, indicating some minor differences between complexes

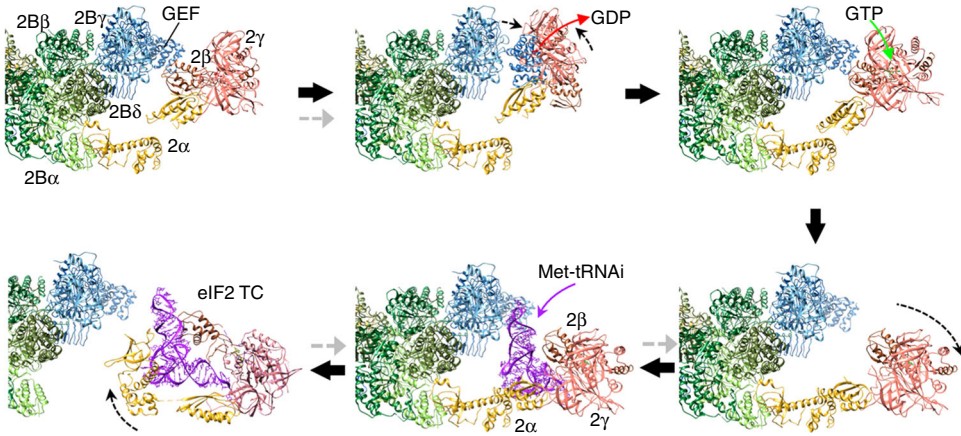

**Fig. 6** Model for eIF2B GDP exchange and ternary complex (TC) formation. A series of steps for guanine nucleotide exchange factor (GEF) action and recruitment of initiator tRNA to form TC based on our structures and docking of the GEF domain (PDB 1PAQ), GDP (PDB 4RD6) or initiator tRNA (3JAP, chain 1), as outlined in the main text and Supplementary Movie 2. Grey arrows indicate steps affected by eIF2(αP)

Based on our structures we have developed a scheme for GEF action and Met-tRNAi binding to eIF2/eIF2B (Fig. 6, Supplementary Movie 2). Here eIF2 is anchored to eIF2B via the strong eIF2α interaction. This fits with previous ideas where eIF2 and eIF2B were suggested to bind initially in a phospho-sensing binding mode[10,20]. eIF2B$\epsilon^{GEF}$ is depicted displacing eIF2β to engage with eIF2γ and release GDP, enabling binding of GTP (Fig. 6, top). Following nucleotide exchange for GTP, eIF2 binds Met-tRNAi to form the TC. We showed recently that eIF2B and Met-tRNAi each compete for eIF2, suggesting that a transient eIF2/TC complex can form[3]. Remarkably and consistent with these findings, we are able to dock Met-tRNAi into the most open conformation of the eIF2/eIF2B complex (Fig. 2g), so that all contacts between the tRNAi and eIF2βγ and eIF2α domain 3 observed in prior structures are maintained. Only eIF2α domain 1 and 2 contacts with tRNAi are replaced with eIF2B contacts[24]. In this position, the tRNAi anti-codon stem projects into a space between eIF2Bγ and eIF2Bε at the catalytic arm (Fig. 6). This modelling suggests that a eIF2B/TC intermediate can form as predicted previously[3]. When combined together, this sequence of events provides a simple model linking nucleotide exchange with TC formation (Fig. 6; Supplementary Movie 2).

While the precise mechanism of how eIF2αP inhibits GEF action is not yet resolved, one possibility consistent with the available data[3] is that phosphorylation stabilises the phospho-loop conformation to permit tighter binding affinity to eIF2B. A failure to release eIF2 therefore locks eIF2αP/eIF2B together and hence sequesters eIF2B (grey arrows in Fig. 6). As eIF2B is always found in limiting concentrations in vivo, a lack of free eIF2B limits TC formation impairing general protein synthesis initiation, but activating the ISR.

Finally, our structures also provide further insight into the mechanism of action of the pox virus inhibitor of PKR, K3L. Pox viruses including variola and vaccinia express K3L proteins that antagonise the action of PKR, preventing PKR-promoted shut down of protein synthesis in infected cells and thereby promoting virus production[45,46]. K3L is a structural mimic of eIF2α domain 1 and shares the conserved KGYID sequence important for both K3L and eIF2α to interact with PKR[37,47] and, as shown above, for eIF2α to also bind eIF2Bα (Supplementary Fig. 8a). Both PKR and K3L have been expressed in yeast cells to study their actions. PKR is toxic, phosphorylating almost all eIF2 in vivo, and this toxicity can be rescued by co-expression of K3L[48]. Growth rescue in this and other cellular contexts implies that K3L can bind and inhibit PKR kinase activity without also binding and inhibiting eIF2B.

Structural alignment of K3L (PDB 1LUZ) onto our eIF2/eIF2B structure reveals steric clashes between K3L$^{V44}$ and K3L$^{K45}$ and eIF2Bα$^{L81}$, as well as between K3L$^{M48}$ and K3L$^{V51}$ and the eIF2Bα carboxy terminal residues Y304 and D305 (Supplementary Fig. 8b) that does not occur upon K3L/PKR docking[37]. In addition, the K3L structure does not make any direct interaction with eIF2Bδ in this model. The surface electrostatic potential here also differs markedly between eIF2α and K3L (Supplementary Fig. 8c). Hence, K3L proteins have evolved to specifically inhibit PKR kinase activity while permitting the eIF2/eIF2B interactions required for productive protein synthesis and hence viral infection to occur. As there has been increased interest in targeting the eIF2/eIF2B regulatory axis of the ISR recently[14], these observations may allow other novel inhibitors to be developed that exploit the differences between kinase and GEF binding to eIF2.

## Methods

**Protein purification.** eIF2 and eIF2B were purified using yeast strains specifically designed to express each complex at a higher level and that lack the yeast eIF2 kinase Gcn2[3,20,25]. Briefly, His$_6$-eIF2 bears a hexahistidine tag at the *GCD11* amino terminus and is expressed in strain GP3511 in the YPD medium[49]. eIF2 was purified from cell extracts made from cell pellets ground under liquid nitrogen in a 6870 Freezer Mill (SPEX SamplePrep) by sequential nickel affinity (Qiagen), HiTrap heparin and HiTrap Q sepharose columns (GE Healthcare). Flag-eIF2B expression strain GP5949 expresses all five eIF2B genes in high-copy plasmids and bears a tandem Flag and a hexahistidine tag at the C terminus of *GCD1* (eIF2Bγ), while a similar strain GP7055 has the hexahistidine tag moved to the C terminus of *GCD6* (eIF2Bε)[25]. Flag-eIF2B was purified from both strains grown in selective SC-Ura-Leu medium[49]. Cells were lysed by grinding under liquid nitrogen in a 6870 Freezer Mill and eIF2B recovered using Flag-M2 affinity agarose (Sigma-Aldrich). Active Flag-PKR was purified from strain GP6065, following growth in the ScGal-Ura-Leu medium[49]. Proteins were stored at −80 °C.

**eIF2/eIF2B complex formation for cryoEM.** For eIF2 phosphorylation, purified PKR and eIF2 (0.08 μg PKR per 10 μg of eIF2) were mixed at room temperature in eIF2 storage buffer supplemented with 0.5 mM ATP, 10 mM MgCl$_2$ and 5 mM NaF, and then incubated for 15 min. The extent of phosphorylation was monitored by SuperSep Phos-tag gel electrophoresis (Fujifilm, Japan) and immunoblotting with polyclonal antibodies to yeast eIF2α (1:1000 dilution)[3]. For complex formation, a six-fold molar excess of eIF2αP or eIF2 was added to freshly purified eIF2B (from 5949 for eIF2αP/eIF2B complexes or GP7055 for eIF2/eIF2B complexes). Proteins were incubated on ice for 10 min and fractionated by size exclusion chromatography with multi-angle light scattering using a Superose 6 10/300GL column (GE Healthcare) in Tris-LS buffer (20 mM Tris-HCl, 100 mM KCl, 1 mM Tris(2-carboxyethyl)phosphine-HCl, pH 7.5) (Supplementary Fig. 1). Protein fractions corresponding to eIF2αP/eIF2B or eIF2/eIF2B complexes (~1 MDa) were pooled and concentrated using centrifugal concentrators (150 kDa molecular weight cut-off (MWCO)). Protein concentration was measured using the Bradford assay (Bio-Rad, Hercules, CA, USA).

**CryoEM grid preparation, data collection and processing**. For initial data collection, 3 μl of a 0.35 mg/ml eIF2αP/eIF2B sample was loaded onto glow-discharged 200 mesh Au Quantifoil R2/2 grids (Electron Microscopy Sciences), blotted for 2 s in FEI Vitrobot Mark III (at 21 °C, 100% humidity) and plunge frozen in liquid ethane. Images were taken on a Titan Krios transmission electron microscope (FEI), operating at 300 kV voltage, equipped with K2 Summit direct electron detector (Gatan) and GIF Quantum energy filter (Gatan). Images were collected using the FEI EPU software at ×37,313 magnification to give a pixel size of 1.34 Å. Images were exposed for 6 s and fractionated into 40 frames, with a dose of 8–10 e−/px/s, yielding a total dose of ~30 e−/Å2. Frames in the images were aligned using dosefgpu driftcorr[50] and a weighted sum from frames 3 to 40 was computed. Defocus parameters were estimated using CTFFIND4[51]. Images were processed using a standard workflow (particle picking, 2D classification and selection, 3D classification and refinement) in RELION 1.4[52].

Processing revealed a strong angular preference of the particles and failed to produce a reasonable 3D reconstruction. Therefore, different sample preparation and data collection procedures were then used to improve the angular distribution of the particles. Three microlitres of a 0.25 mg/ml eIF2αP/eIF2B sample was loaded onto a glow-discharged 400 mesh Cu lacey carbon grid with a 3 nm ultrathin carbon support film (Agar Scientific), blotted for 2 s in FEI Vitrobot Mark III (at 21 °C, 100% humidity) and plunge frozen in liquid ethane. Images were taken on Titan Krios transmission electron microscope (FEI), operating at 300 kV voltage, equipped with K2 Summit direct electron detector (Gatan) and GIF Quantum energy filter (Gatan). Images were taken using 35° stage tilt at ×37,313 magnification, yielding a pixel size of 1.34 Å. Data were collected using the FEI EPU software. Images were exposed for 12 s and fractionated into 48 frames, with a dose of ~5 e−/px/s, yielding a total dose of ~40 e−/Å2.

Movie frames (2–48) were aligned using Motioncor2[53], dividing images in 5 × 5 patches and using dose weighting. Defocus parameters of the images were determined using GCTF v1.06[54]. Initially 2494 particles were manually picked, 2D classified and four best class average images were used as references to automatically pick 249,042 particles in Relion2[55]. Particle set was then manually cleaned by removing obvious false positives (carbon edge, contaminants) and picking any remaining particles, giving a set of 196,242 particles. GCTF v1.06[54] was then used to estimate defocus parameters on a per-particle basis. The set was cleaned by four cycles of 2D classification, where particles belonging 2D classes showing high-resolution features were selected and classified again. The cleaned dataset contained 47,236 particles. Initial processing of smaller in-house datasets showed a homogeneous density with a clear two-fold symmetry in the centre of the structure with weaker lateral densities, which did not follow the symmetry as closely. Initially C2 symmetry was used when focussing on the stable core of the structure. Particles were refined against a reference structure acquired by processing the smaller in-house dataset using ab initio model building in cryoSPARC[56] and filtered to 60 Å. The set was then 3D classified to five classes without image alignment. One class with higher resolution features contained 42,622 particles, which gave a resolution of 4.3 Å after further refinement (using a mask covering the core of the structure).

To improve this further, movies from the first described dataset (200 mesh Au Quantifoil R2/2 grid, no stage tilt during collection) were realigned using Motioncor2[53], dividing images in 5 × 5 patches and using dose weighting. Defocus parameters were determined using GCTF v1.06[54] on a per-particle basis. Particles were re-extracted and 2D classified. Particles in classes showing strongest density of eIF2αP (based on initial model fitting to the preliminary 3D map) were selected and 50,000 best particles (based on MaxValueProbDistribution parameter in Relion2) from this set were added to the 42,622 particles from tilted-stage dataset. The combined particle set was refined and subsequently 3D classified into four classes with no image alignment. A single class containing 64,541 particles displayed high-resolution features, and was refined further and gave an overall resolution for the core of the eIF2αP/eIF2B molecule of 3.9 Å. Local resolution was determined using Resmap[57], and indicated features in the core of the structure were resolved at resolutions up to 3.5 Å. The weaker lateral densities showed local resolution in the range of 8–18 Å (Supplementary Fig. 2g).

Attempts to use classification with no symmetry and various masks to sort and improve the definition and resolution of the variable domains at the sides were not successful and no relationship between domain movements on the two sides could be identified. Therefore, variable domains on each side were analysed independently, using a Localised Reconstruction[58] script in Relion2 to extract and align two half-particles (representing two sides of the molecule) from each of the particles in the original set (Supplementary Fig. 3b). The new half-particle set was then subjected to 3D classification with no image alignments to generate independent classes containing 14–22,000 particles per class of half-particles where eIF2γ is differently orientated relative to eIF2B. Additional classifications into larger number of classes similarly yielded classes with relatively even particle distribution and very slight variations of eIF2αP (domain 3)/eIF2γ conformations without adding further insight into eIF2-eIF2B interactions.

For unphosphorylated eIF2/eIF2B structure determination, eIF2/eIF2B samples (0.28 mg/ml) were prepared using glow-discharged 400 mesh Cu lacey carbon grids with 3 nm ultrathin carbon support film (Agar Scientific), blotted for 1 s in FEI Vitrobot Mark IV (at 21 °C, 100% humidity) and plunge frozen in liquid ethane. Data collection and processing was very similar to the tilted eIF2αP/eIF2B dataset,

except that the initial model was determined using the Relion2.1 initial model program[55].

Images were collected with a defocus target range of 1–3 μm, using the FEI EPU software at ×37,313 magnification to give a pixel size on 1.34 Å. Images were exposed for 14 s and fractionated into 40, 60 or 100 frames, with a dose of ~5 e−/px/s[1], yielding a total dose of ~34 e−/Å2. Frames in the images were aligned using Motioncor2 dividing images in 5 × 5 patches and using dose weighting. Weighted sums from all frames except the first 2 were computed. GCTF was used to determine the CTF parameters, per micrograph initially, and later refined on a per-particle basis.

In Relion 2.1[55], 2024 images were selected for particle picking. Initially, 3090 particles were manually picked. These were classified into 20 2D classes. The best eight classes (containing 2162 particles) were selected and used as references for automated particle selection. Particles on the lacey carbon, not in holes, were manually deleted in Relion, before extraction and further iterative cleaning by 2D classification, resulting in an initial set of 114,390 particles.

An initial model was determined using Relion2.1, applying two-fold symmetry. After three further rounds of 2D classification, a set of 46,064 was obtained, and 3D classified into four 3D models, each with a highly similar overall conformation (initial model filtered to 60 Å, no masks applied, reconstruction radius 320 Å, two-fold symmetry applied). One class containing 23,274 particles had significantly higher resolution features, and the corresponding particle set and model were subject to automated 3D refinement, using a mask defining the core of the molecule, as before for the eIF2αP/eIF2B structure. The final 3D map was refined to an overall resolution of 4.6 Å. Local resolution analysis[59] indicates a higher resolution of 4.1 Å in the central core region, while the resolution of the map at the periphery (containing the flexible eIF2 arm features) is ~11–18 Å (Supplementary Fig. 6c).

**Atomic model building and refinement**. To build atomic model of eIF2B into the core of the eIF2αP/eIF2B structure, a homology model of *S. cerevisiae* eIF2B was made using Modeller[60] and the crystal structure of *S. pombe* (PDB 5B04) as a reference[17]. Subunits of the homology model were then individually fitted into the eIF2αP/eIF2B map using UCSF Chimera[61]. The eIF2αP/eIF2B map did not show any density for the β-helical domains of eIF2Bγ subunit (residues 416–578); therefore, these were deleted from the homology model. In addition to the eIF2B subunits, our core map had density for domains 1 and 2 of eIF2α. For model building, eIF2α from a cryoEM structure of 48S preinitiation complex (PDB 3JAP, residues 3–174)[24] was fitted into our map using UCSF Chimera[61].

The eIF2αP/eIF2B atomic model was then refined in Phenix.real_space_refine[62] using global minimisation, simulated annealing, B-factor refinement and non-crystallographic symmetry (NCS) restraints. The model was then manually adjusted in COOT[63] and refined (parameters as above) several times. Final model statistics were generated using MolProbity[64].

For eIF2/eIF2B modelling, the atomic model of the eIF2αP/eIF2B core [eIF2B with eIF2α (3–174)] was rigid body fitted to the map of eIF2/eIF2B in UCSF Chimera[61], refined in Phenix.real_space_refine[62] using global minimisation, B-factor refinement and NCS restraints, and manually adjusted in COOT[63]. After several cycles of refinement and manual adjustment, final model statistics were generated using MolProbity[64].

The weak lateral densities were assigned to eIF2 and by fit eIF2 subunit atomic models (α 182–265, β 127–143 and γ 98–519) from a cryoEM structure of the partial yeast 48S preinitiation complex (PDB 3JAP)[24]. These were fitted as rigid bodies into the eIF2αP/eIF2B and eIF2/eIF2B maps (Fourier filtered to 15 Å resolution) using UCSF Chimera[61].

**Coordinate and map comparisons**. FSCs for map resolution determination were calculated using the gold-standard method[65], which compares two independently refined halves of the dataset (implemented in Relion). The resolutions reported are based on the FSC = 0.143 criterion. As a further check against overfitting during map refinement, Relion also calculates the FSC from two half-maps independently refined against the model using a copy of the images with phases randomised beyond a high-resolution threshold. These are labelled phase randomised half-maps in the figures. Signal in these beyond the threshold resolution would indicate spurious high-resolution features and correlation introduced in the refinement process, for example, by sharp masks[66].

Atomic model building and fitting were done into the final postprocessed maps produced in Relion. The program Locscale[59], as implemented in the CCP-EM suite[67], was used to rescale the maps so that the density of the weaker and poorer resolution peripheral features was of comparable strength when visualised. When making visual comparisons of the two maps (e.g. Supplementary Fig. 6), the scaling and resolution of the higher resolution eIF2αP/eIF2B map was matched to the lower-resolution one, using the program 3Dradamp to modify the reciprocal space 3D radially averaged amplitude profile of one structure to the other, as before[68]. This procedure makes map features directly comparable and unrelated to resolution or amplitude profile differences. Alignment of atomic structure coordinates and RMSD calculations were done with the program Gesamt[69], and part of the CCP4 suite[70] using default *High* model parameters. Values reported are either pairwise comparisons of aligned and matched Cα atoms or RMSD compared to a consensus structure produced from a simultaneous multi-structure optimal alignment.

**Map and model validation.** Cross-validation, as in Fernandez et al.[71], was used to show that the model to map weight during atomic model refinement was chosen correctly to avoid overfitting. The positions of all atoms in the penultimate model were randomised with a mean displacement of 0.5 Å using the Phenix shake command. Then, this model was refined against one of the independently refined half-maps (from the final EM structure) from Relion. A density map was made from the PDB coordinates using the program Makedensity2, available as part of the DockEM package[72] in the CCP-EM suite. FSC$_{WORK}$ is the correlation of this model with the half-map it was refined against. FSC$_{TEST}$ is the FSC of the refined model against the other, unseen, half-map. Close correspondence between curves indicates the absence of overfitting in the model refinement. FSCs were calculated using Relion_image_handler.

**Yeast suppressor genetic analysis.** To identify mutants in eIF2α that suppress the 3-amino-1,2,4-triazole-sensitive phenotype of a gcd2-E381Q strain, a plasmid encoding SUI2 (yeast eIF2α) was randomly mutated using XL1-Red mutator strain of Escherichia coli (Agilent Technologies), and the resulting mutant plasmid pool was transformed into gcd2-E381Q yeast and resulting strains were screened for 3AT resistance. sui2-I63N was the only allele found (following plasmid rescue, retransformation and DNA sequence analysis) to confer 3AT resistance to gcd2-E381Q cells. The genetic analysis shown in Supplementary Fig. 5 was performed in strains derived from GP3428 (gcd2Δ sui2Δ) and GP4346 (gcn3Δ sui2Δ)[27] where the indicated allele for each factor is the sole source of that protein. Immunoblotting of whole-cell extracts used rabbit polyclonal antibodies to yeast eIF2α (1:1000 dilution) and phospho-specific eIF2αP (Cell Signalling Technologies #9721, 1:1000 dilution)[44].

**eIF2B-PKR competition assay.** Ten microlitres of reactions contained eIF2 (1 μM) and PKR (10 nM), and eIF2B (0–2 μM) diluted in 30 mM HEPES, pH 7.5, 100 mM KCl and in the presence of 10 mM MgCl$_2$, 0.5 mM ATP, and bovine serum albumin (3 μM). Following incubation for 20 min at room temperature, reactions were stopped by the addition of sodium dodecyl sulfate-polyacrylamide gel electrophoresis loading dye and incubation at 95 °C for 5 min. eIF2 phosphorylation status was visualised by separating the reaction samples on a SuperSep Phos-tag acrylamide gel (Fujifilm) and western blotting probing with anti-eIF2α antibodies (1:500) and detected by IRDye® 800CW Donkey anti-chicken IgG-labelled secondary antibodies (Li-Cor 32218, 1:10,000). Signals were quantified with the Image Studio software (Li-Cor). The source data underlying Fig. 4a are provided within the Source Data file.

**Reporting summary.** Further information on research design is available in the Nature Research Reporting Summary linked to this article.

## Data availability

The cryoEM density maps and atomic coordinates have been deposited in the Electron Microscopy Data Bank and the Protein Data Bank, under accessions EMD-4404 and 6I3M, respectively, for the eIF2αP/eIF2B complex and EMD-4428 and 6I7T for the eIF2/eIF2B complex. The source data underlying Fig. 4a and Supplementary Figs. 1a–c, 2f, 5, and 6d are provided as a Source Data file.

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

## Acknowledgements

We acknowledge the University of Manchester Faculty of Biology, Medicine, and Health EM core facility for staff and instruments to screen and analyse samples and the Wellcome Trust for equipment grant support to the EM Core Facility, as well as the University of Manchester Computational Shared Facility for computing resources. In addition, we are grateful to Diamond Light Source (Oxfordshire, UK) for access and support of the CryoEM facilities at the UK National Electron Bio-Imaging Centre (eBIC), (proposal EM15436, EM16619), funded by the Wellcome Trust, MRC and BBSRC. This work was supported by BBSRC grants BB/L020157/1 to G.D.P. and A.M.R., as well as BB/N014049/1 and BB/M006565/1 to G.D.P., as well as a BBSRC doctoral training program grant (BB/M011208/1) to the University of Manchester.

## Author contributions

T.A., M.G., Y.Z., and M.D.J. performed protein purification and biochemical analyses. Z.L. performed the genetics analysis. T.A. determined the structure of eIF2αP/eIF2B. M.G., A.M.R., and T.A. determined the structure of eIF2/eIF2B. M.G., A.M.R., and T.A. performed structure validations and all authors analysed the data. A.M.R. and G.D.P. conceived and led the study, analysed data, and co-wrote the manuscript with assistance from T.A. and M.G. All authors approved the final manuscript.

## Additional information

**Competing interests:** The authors declare no competing interests.

**Journal Peer Review Information:** *Nature Communications* thanks Madhusudan Dey, Felix Weis and the other, anonymous, reviewer(s) for their contribution to the peer review of this work. Peer reviewer reports are available.

