## [Peer Review File · Nature Communications]

Reviewers' comments:

Reviewer #1 (Remarks to the Author):

This study presents the first high-resolution structures of eIF2, phosphorylated and unphosphorylated, bound to eIF2B. The results described here, in accordance with a wealth of previous genetic and biochemical data, provide decisive new insights into the mechanism of nucleotide exchange and activation of eIF2. These findings are important not only for the translation initiation field, but also for a larger audience as this step is a key event of the integrated stress response and cell homeostasis.

- The authors state that eIF2B cellular levels are lower than eIF2 and described also their in vitro formed complexes with two molecules of eIF2 bound to one molecule of eIF2B. It could be useful for the general audience to comment the importance and the pertinence of such a ratio in an in vivo and/or physiological context.
- The combination of tilted and untilted data was instrumental to obtain a high-resolution and high-quality cryo-EM map of the eIF2 α P/eIF2B complex: Did the authors use the method described in Naydenova et al., Nat Commun (2017) to decide the tilt angle and the ratio of tilted and untilted data? Why did they use only tilted data for the eIF2/eIF2B complex?
- The study and description of eIF2 γ arm flexibility is a key point of this manuscript, unfortunately the methods used to achieve such a classification is only succinctly described. From a technical point of view it is important to describe more thoroughly in the methods section the corresponding processing together with a new supplementary figure describing the classification scheme.
- In the context of ~ 4 Å resolution maps, the validation of the atomic model refinement parameters is of prime importance. The authors needs to include FSC curves of the models against the maps as well as the half-maps validation FSC curves as described in Fernandez et al., Cell (2014) to demonstrate the absence of overfitting in their models.

Minor comments:

- The authors are sometimes using two numbers after the decimal point for RMSD and resolution; they should only use one.
- eIF2 α P is sometimes written eIF2P (Supplementary Table 3) or eIF2 (Supplementary Figure 1 panel c).
- Are the SEC-MALLS traces different for eIF2 α P/eIF2B and eIF2/eIF2B complexes? If so they both need to be included.
- For clarity and to allow easier comparison, Supplementary Figures 2 and 6 (describing the cryo-EM structures) need to be similar: similar 2D classes, same scale for the local resolution maps, same orientation for the angular plots, etc...

Reviewer #2 (Remarks to the Author):

The decameric eIF2B is the guanine nucleotide exchange factor (GEF) for eIF2, a trimeric GTPase that brings the initiator Met-tRNA_i to the ribosome. eIF2B is one of the main targets of regulation of protein synthesis. The substrate eIF2 is phosphorylated by several stress-induced kinases, in what is collectively known as the Integrated Stress Response (ISR). Phosphorylated eIF2 (eIF2(a-P)) acts as a competitive inhibitor of eIF2B. Dysregulated ISR is implicated in a number of neurodegenerative disorders, including Alzheimer's Disease. The mechanisms of eIF2B action and regulation are of great

scientific and medical importance and are currently the subject of high interest and intensive research by multiple labs in the field. In fact, currently three other manuscripts reporting the Cryo-EM structures of eIF2B:eIF2 (enzyme:substrate) and/or eIF2B:eIF2(a-P) (enzyme:inhibitor) complexes have been deposited pre-publication in BioRxiv.

This manuscript reports the Cryo-EM structures of the eIF2B:eIF2(a-P) complex and the eIF2B:eIF2 complex. A mechanism for eIF2B action and regulation by phosphorylation of eIF2 is proposed. However, the eIF2B:eIF2 complex (enzyme-substrate) structure presented here is in stark contrast to that reported in two manuscripts, from the Walter and Ito labs, deposited with BioRxiv. The work from the Ito lab is especially of note because it reports the structures of both the eIF2B:eIF2 (enzyme:substrate) and/or eIF2B:eIF2(a-P) (enzyme:inhibitor) complexes, and they are extremely different: essentially, mirror images along the two-fold symmetry axis of eIF2B.

Previously published cross-linking (Kashiwagi et al., 2016) are quantitatively, but not qualitatively different between the eIF2B:eIF2(a-P) and eIF2B:eIF2 complexes, and are not fully consistent with any one structure, but are consistent with a combination of both structures. Therefore, the most likely explanation is that the eIF2B:eIF2 complexes exist in an equilibrium between two drastically different states, and that phosphorylation shifts the equilibrium toward one of them. Accordingly, the eIF2B:eIF2 complex structure reported by the Walter and Ito labs is consistent with those crosslinks from Kashiwagi et al. that are weaker in the phosphorylated complex, as expected, whereas the eIF2B:eIF2 complex structure reported here, like the eIF2B:eIF2(a-P) complex structure, is not.

While I cannot know why the eIF2B:eIF2 structure presented here differs from those reported by the Walter and Ito labs, I suspect it could be due to the initial approach of particle selection employed here, the Relion initial model program. The authors say they started with ~3000 manually picked images, which could have skewed the automatic selection toward the complexes similar to the reference eIF2B:eIF2(a-P) complex, potentially the minor species in the sample.

To be clear, ordinarily, the assumption that the active and inhibited complexes of eIF2B:eIF2 are similar would be a very solid one (and everyone in the field has shared it for decades), if it weren't for the two newly-solved structures, which show otherwise, and which explain plethora of experimental data.

Therefore, the authors should reevaluate their particle selection approach for the eIF2B:eIF2 complex, explore the possibility that a second species exists in their samples (possibly even the predominant species), as well as report what % of the particles were used for structure determination, what other classes exist in the sample, what they correspond to, and what % of the particles they represent. Without this additional analysis, any discussion of the structure of eIF2B:eIF2 and the mechanism of catalysis is premature. After reassessing the data, the authors should either modify their conclusions accordingly or discuss the possible reasons for the discrepancies in structures, as well as the scientific implications.

Reviewer #3 (Remarks to the Author):

During translation initiation, the GTP-bound form of eukaryotic initiation factor 2 (eIF2) delivers the initiator methionyl-tRNA (Met-tRNA^{Met}) to the 40S ribosomal subunit and then dissociates as a GDP-bound form. GDP-eIF2 is then recycled into GTP-eIF2 by the guanine nucleotide exchange factor (GEF) eIF2B for assembly of a new ternary complex (TC) with Met-tRNA^{Met} for the next round of translation initiation.

Under various stressful conditions, cells reprogram their translational machineries, partly by activating an integrated stress response (ISR) that leads to phosphorylation of the α -subunit of eIF2 on the residue Ser51. Phosphorylated eIF2 (eIF2 α P) has been known to bind to eIF2B very tightly, thus reducing its nucleotide exchange activity and consequently reducing the reassembly of the TC. The reduced level of TC decreases the average rate of translation; however, increases the rate of translation of a specific subset of mRNAs in order to adapt to stress. A large amount of genetic and biochemical data support the functional interaction between eIF2 and eIF2B; however, the structural data have been lacking.

Both eIF2 and eIF2B are multi-subunit proteins. eIF2 consists of three α , β and γ , whereas eIF2B consists of five α , β , γ , δ and ϵ . A few structures of eIF2 (e.g., PDB ID = 3JAP) and eIF2B (e.g., PDB IDs = 5B04, 6CAJ and 6EZO) are known. The crystal structure of eIF2B of the fission yeast *S. pombe* reveals that two copies of each of these subunits assemble into a decamer (PDB ID = 5B04). Like the fission yeast eIF2B, the cryo-EM structure of human eIF2B has also shown to adopt a decamer in the presence of a small molecule ISRIB (PDB ID = 6CAJ and 6EZO). In this manuscript, authors have resolved two cryo-EM structures of eIF2B purified from the budding yeast *S. cerevisiae*: one bound to phosphorylated eIF2 (Fig 1) and other to non-phosphorylated eIF2 (Fig 5). They have observed that the subunit structures of the budding yeast eIF2B in the complex of eIF2B/eIF2 α or eIF2B/eIF2 α -P are very similar to the published decameric structure of the fission yeast eIF2B or human eIF2B. However, their structures provide direct evidence that the eIF2 α subunit contacts mainly with the regulatory α and δ subunits of eIF2B (Supplemental Fig 4).

Moreover, the complex of eIF2B/eIF2 or eIF2B/eIF2 α -P reveals that the Ser51 phosphorylation site and the KGYID motif of eIF2 α make the core contacts with the eIF2B α subunit (Fig 3 and Supplemental Fig 5). These observations are consistent with the prior molecular genetics and biochemical studies that suggest that the residues surrounding the Ser51 phosphorylation site and the KGYID motif of the eIF2 α interact with the eIF2B complex. To provide further genetic evidence for their structure, authors analyze the interacting surfaces of both eIF2 α and eIF2B α , mutate a few surface residues, and show that a single mutation in eIF2 α (e.g., I63N) or eIF2B α (e.g., T41A) exhibited a GCN⁻ phenotype (Supplementary Fig 5). Finally, authors propose a model on how eIF2B catalyzes the exchange of GTP for GDP. They propose that binding of eIF2 α to eIF2B triggers a conformational change in the D3 domain of eIF2 α bound to eIF2B $\beta\gamma$ complex, α moving away from the catalytic eIF2B $\beta\gamma\epsilon$ subunits (Fig 2 and Fig 6). These studies significantly advance our knowledge on translational control by eIF2 α phosphorylation.

Some minor comments:

1. This reviewer had to read a couple of times to understand the Fig 2. The legend of Fig 2 should be described clearly. How did they align two structures? What is the reference? Difficult to understand the lines - "aligned onto eIF2 α domain 3 TC from 3JAP"... b-g "aligned modelled onto 3JAP eIF2 (as in panel a)"...
2. Authors suggested an elbow-like rotation of eIF2 α domains 2 and 3 when eIF2B binds to eIF2 (Fig 2 and Fig 6). This reviewer was wondering two things. (I) Did authors try to resolve the structure of eIF2B without its α -subunit? (II) How can it be explained the observation that the eIF2B γ arm is relatively far apart from the eIF2 α domain 3 in the complex of eIF2B/eIF2?
3. Line 227: eIF2B α -T41 and eIF2B α -E44 approach eIF2 α -Y82. What do authors mean by stating the word "approach"? What's the distance between residues eIF2B α -T41 and eIF2 α -Y82 or residues eIF2B α -E44 and eIF2 α -Y82.

4. The phosphorylated side chain of the residue eIF2 α -Ser51 appears to be inside the structure (Supplemental Fig 5C). If that is correct, how will you connect the physiological relevance of Ser51 phosphorylation?

5. In Fig 4, authors show that PKR and eIF2B compete for binding to eIF2 α . Authors also show that Ser51 phosphorylation of eIF2 α -I63N mutant protein by GCN2 was not affected (Supplemental Fig 5b), but the eIF2 α -I63N mutant eliminated the GCN2-mediated translational control when eIF2B α was absent or mutated at the residue eIF2B α -T41 or α -F73 (3-AT sensitive phenotype, Supplemental Fig 5a). Does the eIF2B complex compete with PKR when it carries an eIF2B α -T41A mutation or when the complex is devoid of the eIF2B α ?

6. The discussion on the viral translational control and the PKR kinase inhibition by viral protein K3L in the last section are highly speculative and based on the observation upon superposition of K3L structure on eIF2 α in the complex of eIF2B (Supplemental Fig 8). Please comment on how would you interpret observation that the corresponding yeast eIF2B α residues Y304 and D305 are absent in its human ortholog when aligned them.

Reviewer 1

This study presents the first high-resolution structures of eIF2, phosphorylated and unphosphorylated, bound to eIF2B. The results described here, in accordance with a wealth of previous genetic and biochemical data, provide decisive new insights into the mechanism of nucleotide exchange and activation of eIF2. These findings are important not only for the translation initiation field, but also for a larger audience as this step is a key event of the integrated stress response and cell homeostasis.

We thank the reviewer for their thoughtful and helpful review of our manuscript.

- The authors state that eIF2B cellular levels are lower than eIF2 and described also their in vitro formed complexes with two molecules of eIF2 bound to one molecule of eIF2B. It could be useful for the general audience to comment the importance and the pertinence of such a ratio in an in vivo and/or physiological context.

In the introduction paragraph 2 we state:

As eIF2B levels are lower than eIF2 in cells, partial phosphorylation is sufficient to attenuate protein synthesis initiation¹.

We have now added a sentence to the end of the first results section to make the point that the reviewer is suggesting.

As in cells eIF2B levels are limiting, the 1:2 eIF2B:eIF2 structure is fully consistent with partial eIF2 phosphorylation in vivo being able to fully repress eIF2B function and protein synthesis initiation.

- The combination of tilted and untilted data was instrumental to obtain a high-resolution and high-quality cryo- EM map of the eIF2 α P/eIF2B complex: Did the authors use the method described in Naydenova et al., Nat Commun (2017) to decide the tilt angle and the ratio of tilted and untilted data?

We did not use the method described by Naydenova & Russo (2017) to choose the tilt angle, because the paper was not published when we collected our eIF2 α P/eIF2B complex datasets. We did geometrical modelling to choose the angle of 35 degrees, while taking into account that data quality drops with higher tilt due to increase in ice thickness, and sample movement or charging. This modelling showed that 35 degrees was sufficient to create the angular sampling required. We chose the optimal amount of zero degree data to merge with the titled images empirically, so as not to bias the procedure by over representation of these preferential views.

Subsequent analysis using the program of Naydenova & Russo (2017) suggests an angle of 38 degrees and collecting ~2.9 times as much tilted data as at zero. We used 2:1 tilted to non-tilted data, by restricting the amount of zero degree data included.

-Why did they use only tilted data for the eIF2/eIF2B complex?

For our second structure, eIF2/eIF2B (not phosphorylated), we had established that the tilt of 35 degrees was a good choice giving sufficient views to determine the structure. We also had established that inclusion of additional zero degree data would likely result in only a small improvement. As we had time limited access to the microscope at the Diamond national EM facility, we focused our efforts on obtaining the tilted images.

-The study and description of eIF2 γ arm flexibility is a key point of this manuscript, unfortunately the methods used to achieve such a classification is only succinctly described. From a technical point of view it is important to describe more thoroughly in the methods section the corresponding processing together with a new supplementary figure describing the classification scheme.

We have added a new figure (Supplementary Fig. 3b), as suggested by the reviewer and modified both the methods description and the main results description accordingly to clarify how this analysis was done.

-In the context of ~4 Å resolution maps, the validation of the atomic model refinement parameters is of

prime importance. The authors need to include FSC curves of the models against the maps as well as the half-maps validation FSC curves as described in Fernandez et al., Cell (2014) to demonstrate the absence of overfitting in their models.

We have added the model vs. map FSCs to supplementary figures 2 and 6.
We have added FSC cross-validation plots to show that the model-weighting in refinement was correct to avoid overfitting.

In addition, we have added an extra plot “Phase randomised half maps” to the map FSC figure to indicate absence of bias and over fitting in the density map refinement.

Relevant explanations to the figure legend and method sections have been added.

Minor comments:

- The authors are sometimes using two numbers after the decimal point for RMSD and resolution; they should only use one.

We have removed the second decimal place where this appeared in the main text.

- eIF2 α P is sometimes written eIF2P (Supplementary Table 3) or eIF2 (Supplementary Figure 1 panel c). This has been corrected. The SEC-MALLS traces for both eIF2 α P/eIF2B and eIF2/eIF2B are now shown in Supp. Fig 1C.

- Are the SEC-MALLS traces different for eIF2 α P/eIF2B and eIF2/eIF2B complexes? If so they both need to be included.

The SEC-MALLS traces are identical for both complexes.

- For clarity and to allow easier comparison, Supplementary Figures 2 and 6 (describing the cryo-EM structures) need to be similar: similar 2D classes, same scale for the local resolution maps, same orientation for the angular plots, etc...

Similar 2D classes: The images and plot data shown were generated automatically by the software used (Relion), rather than selected by us. Hence the apparent rotation of the particle classes. The class averages are ranked in order by number of images assigned to the class. To address the point made, we have added a further panel to Supplementary Figure 6a showing selected 2D classes from the Supplementary Figure 2a, arranged in the same order as the eIF2B/eIF2 classes are.

Same scale for the local resolution maps: We have changed the resolution scale shown on the map in Supplementary Figure 6c so that it matches Supplementary Figure 2g.

Same orientation for the angular plots. We have changed the plot shown in Supplementary Figure 6b. In the original plot Relion had placed the molecule randomly on the axis of symmetry.

Reviewer 2

The decameric eIF2B is the guanine nucleotide exchange factor (GEF) for eIF2, a trimeric GTPase that brings the initiator Met-tRNA_i to the ribosome. eIF2B is one of the main targets of regulation of protein synthesis. The substrate eIF2 is phosphorylated by several stress-induced kinases, in what is collectively known as the Integrated Stress Response (ISR). Phosphorylated eIF2 (eIF2(α -P)) acts as a competitive inhibitor of eIF2B. Dysregulated ISR is implicated in a number of neurodegenerative disorders, including Alzheimer's Disease.

The mechanisms of eIF2B action and regulation are of great scientific and medical importance and are currently the subject of high interest and intensive research by multiple labs in the field. In fact, currently three other manuscripts reporting the Cryo-EM structures of eIF2B:eIF2 (enzyme:substrate) and/or eIF2B:eIF2(α -P) (enzyme:inhibitor) complexes have been deposited pre-publication in BioRxiv.

This manuscript reports the Cryo-EM structures of the eIF2B:eIF2(α -P) complex and the eIF2B:eIF2 complex. A mechanism for eIF2B action and regulation by phosphorylation of eIF2 is proposed. However, the eIF2B:eIF2 complex (enzyme-substrate) structure presented here is in stark contrast to that reported in two manuscripts, from the Walter and Ito labs, deposited with BioRxiv. The work from the Ito lab is

especially of note because it reports the structures of both the eIF2B:eIF2 (enzyme:substrate) and/or eIF2B:eIF2(α -P) (enzyme:inhibitor) complexes, and they are extremely different: essentially, mirror images along the two-fold symmetry axis of eIF2B.

Previously published cross-linking (Kashiwagi et al., 2016) are quantitatively, but not qualitatively different between the eIF2B:eIF2(α -P) and eIF2B:eIF2 complexes, and are not fully consistent with any one structure, but are consistent with a combination of both structures. Therefore, the most likely explanation is that the eIF2B:eIF2 complexes exist in an equilibrium between two drastically different states, and that phosphorylation shifts the equilibrium toward one of them. Accordingly, the eIF2B:eIF2 complex structure reported by the Walter and Ito labs is consistent with those crosslinks from Kashiwagi et al. that are weaker in the phosphorylated complex, as expected, whereas the eIF2B:eIF2 complex structure reported here, like the eIF2B:eIF2(α -P) complex structure, is not.

While I cannot know why the eIF2B:eIF2 structure presented here differs from those reported by the Walter and Ito labs, I suspect it could be due to the initial approach of particle selection employed here, the Relion initial model program. The authors say they started with ~3000 manually picked images, which could have skewed the automatic selection toward the complexes similar to the reference eIF2B:eIF2(α -P) complex, potentially the minor species in the sample.

To be clear, ordinarily, the assumption that the active and inhibited complexes of eIF2B:eIF2 are similar would be a very solid one (and everyone in the field has shared it for decades), if it weren't for the two newly-solved structures, which show otherwise, and which explain plethora of experimental data.

Therefore, the authors should reevaluate their particle selection approach for the eIF2B:eIF2 complex, explore the possibility that a second species exists in their samples (possibly even the predominant species), as well as report what % of the particles were used for structure determination, what other classes exist in the sample, what they correspond to, and what % of the particles they represent. Without this additional analysis, any discussion of the structure of eIF2B:eIF2 and the mechanism of catalysis is premature. After reassessing the data, the authors should either modify their conclusions accordingly or discuss the possible reasons for the discrepancies in structures, as well as the scientific implications.

We thank the reviewer for their thoughtful assessment. The reviewer has clearly seen the pre-publications just deposited in BioRxiv within a day or two after our own submission. Obviously, these data were not in the public domain when we wrote our manuscript, precluding each set of authors from discussing the others work. Although we appreciate that the reviewer '*cannot know why the eIF2B:eIF2 structure presented here differs from those reported by the Walter and Ito labs*'. We also do not know the reason(s) for the differences seen. The reviewer suggests that there may be technical reasons for differences that we address below, along with suggesting some alternative ideas.

First, we compare these unpublished studies as far as we are able, with the limited information currently available.

eIF2 α P/eIF2B

We report *S. cerevisiae* eIF2 α P/eIF2B as does a study by Ramakrishnan and colleagues. Ito and co-workers report a very similar cryoEM structure for human eIF2 α P/eIF2B as well as a crystal structure of *S. pombe* eIF2B with *S. cerevisiae* phosphorylated eIF2 α . Walter and colleagues report a cryo EM structure of human eIF2B with phosphorylated human eIF2 α . In all these structures there is remarkable agreement into the binding interface between eIF2B α and δ and eIF2 α .

eIF2/eIF2B

We report *S. cerevisiae* eIF2/eIF2B. Ito and co-workers report a cryoEM structure for human eIF2/eIF2B as well as a crystal structure of *S. pombe* eIF2B with *S. cerevisiae* unphosphorylated eIF2 α . Walter and colleagues report a cryo EM structure of human eIF2B/eIF2 in the presence of the compound ISRIB. Both human structures resolve a structure with only one eIF2 trimer bound to the eIF2B decamer, Walter's study also sees a second complex with two eIF2's bound

to eIF2B, both bound in the same alternate conformation. Ramakrishnan and colleagues do not show eIF2/eIF2B.

Comparisons:

1. Our structure appears highly similar to the of the crystal structure of *S. pombe* eIF2B complexed with *S. cerevisiae* unphosphorylated eIF2 α as reported in the new BioRxiv submission by Ito and colleagues. Their figure 3 panels E and F of the interactions of eIF2 α with eIF2B appears identical to ours. Arguably, this crystal structure is the most comparable with our structure, as both contain *S. cerevisiae* eIF2 α and a yeast eIF2B. Both our structure and this crystal structure also support the conclusions of Ito's own previously published cross-linking experiments that the reviewer refers to (Kashiwagi et al 2016). In that study Ito identified 6 crosslinks between eIF2B α and eIF2 α , 4 between eIF2B β and eIF2 α and 3 between eIF2B δ and eIF2 α . Of these the only altered by eIF2 phosphorylation were 2 eIF2B β crosslinks. The eIF2B α and δ crosslinks were entirely unchanged, in agreement with our structure and Ito's new crystallography data.
2. As the reviewer states our eIF2B/eIF2 and Ito's eIF2B/eIF2 α structures differ from Walter's and Ito's human eIF2B/eIF2 structures. These structures suggest an alternative binding mode is possible. This is clearly both significant and confusing. However, at odds with the reviewer's comment, it is this second conformation seen by Ito and Walter that does not agree with Ito's prior cross-linking (albeit with the *S. pombe* proteins). In their new structures there appears to be no contact between eIF2B α and eIF2 α in the unphosphorylated complex.

These differences suggest either methodological differences or interesting biological differences between species, or alternative intermediate states in the nucleotide exchange process.

Methodological differences. Sample preparation.

Our samples were prepared from proteins rapidly extracted from yeast cells and were processed without any cross-linker and without any detergent. The eIF2/eIF2B interaction is stable to SEC MALLS fractionation. We have likely isolated the most stable conformation. Walter's manuscript states they used an *E. coli* system to express and purify human eIF2B as well as a system that we developed to express human eIF2 in yeast. Their purified proteins were crosslinked with the amine specific BS3 to form complexes that also contain ISRIB. Ito's pre-print manuscript is missing the entire methods section, but in his text he says he also used *E. coli* expression systems for his proteins and added a detergent in the buffer used for complex formation. It is not possible at this point in time for anyone to know the impact of these sample source and preparation differences on the final complexes observed.

Particle selection.

The reviewer is concerned that we may have biased our particle picking and inadvertently missed particles in an alternative conformation:

The authors say they started with ~3000 manually picked images, which could have skewed the automatic selection toward the complexes similar to the reference eIF2B:eIF2(α -P) complex, potentially the minor species in the sample

We wish to assure the reviewer that this is not the case. We made sure that initial selection and the autopicking process selected all potential eIF2/eIF2B complexes visible on our grids for the subsequent steps in our analyses. So we did not select a minority of particles to process.

Example images from the eIF2B/eIF2 grid squares are shown below.

In light of the reviewer's comments we have also examined our data analysis pipeline and see no evidence for an alternative conformation equivalent to that proposed by Walter and Ito in our 2D or our 3D classes.

Initial auto picking gave ~250k potential particles that were reduced to 114k after removing those on the carbon and initial cleaning by 2D classification for images not containing particles, or carbon edges. The set of 114k particles was reduced to 46k in a three further rounds of cleaning by 2D classification. At this stage non-particle features or aggregates were removed. The 46k particle set was subject to 3D classification into 4 classes. These looked similar overall, however one had significantly higher resolution, and contained the most particles. The images were assigned to the 3D classes in the ratio 17%, 18%, 16%, 49%. The higher resolution structure and particle subset (49% class) consisting of 23k particles was selected for further 3D refinement. The 3 rejected 3D classes were of similar overall conformation to the selected class, albeit at significantly poorer resolution, indicative of flexibility in the complex seen in the 2D classes and in the modeling and analysis of the 3D structures. This is shown in the figure below, where all four classes are shown and our atomic model is superimposed on one of the classes to demonstrate they all have a similar overall conformation with eIF2 α density adjacent to eIF2B α and no unexplained density. The remaining domains of eIF2 are mobile and not seen in these representations as their density is weaker.

None of the 2D classes or 3D models indicated either asymmetric one sided 1:1 eIF2:eIF2B complexes or classes approximating to the structure reported by Ito's or by Walter's labs.

We conclude that the absence of a structure similar to Ito's or Walter's data is not due to any bias imposed by our data analysis pipelines.

Biological Differences.

Given the remarkable similarity of the overall structures of isolated eIF2 and eIF2B from different sources and the similarities of the inhibited eIF2 α P/eIF2B complexes it is obviously highly unexpected that the new studies find different structures. Indeed the reviewer speculates: '*Therefore, the most likely explanation is that the eIF2B:eIF2 complexes exist in an equilibrium between two drastically different states, and that phosphorylation shifts the equilibrium toward one of them.*'

This may be true for the human proteins and it is proposed by the Ito and Walter studies. However we have no evidence to support this idea. Moreover, our model for nucleotide exchange shows that such a large re-arrangement is not necessary to permit both nucleotide exchange and tRNA recruitment. Ramakrishnan and colleagues also propose a similar model to ours.

One contributing element to the high affinity between yeast eIF2 α and eIF2B, that we report in our manuscript, is the complementary surface charge between the binding interfaces. To explore this idea further we have now compared the Coulombic potentials of currently published structure pdb files with our own data.

eIF2 α

When the eIF2B-interacting surfaces of eIF2 α from yeast and human sources are compared they show a similar pattern of positive and negative patches (Figure below, bottom left-rotated to show the interacting surface and with the position of the three consecutive arginines following the regulatory serine indicated).

eIF2B

Also shown are the eIF2B $\alpha\beta\delta$ cores from *S. pombe* eIF2B (left), alongside the two human eIF2B structures with ISRIB bound (from Walter's and Ron's lab, middle) and our eIF2B-eIF2 α core structure (right). Each human structure is missing different and multiple side-chain atoms and some surface loops. Consequently each is missing atoms that would contribute to the calculated surface potentials. Hence they differ considerably from each other and also from those of the eIF2Bs from both yeasts. Despite obvious limitations of these comparisons, there are clear similarities in surface coulombic potential between the *S. cerevisiae* and *S. pombe* surfaces where eIF2 α binds in our structure (circled). Here the coulombic potential is complementary to eIF2 α . This positive v negative contrast is much less evident on the equivalent human eIF2B surfaces. This may suggest, albeit tentatively, that the human proteins may not interact with eIF2 α here with the stability seen between the yeast proteins. However we view this suggestion as too tentative to include in the manuscript text. As we do not have access to the new structure pdb files, we cannot perform comparisons necessary to make any comparisons.

In summary it is our view that the differences observed between studies may be biologically relevant. It is therefore critically important that each study is published promptly so that they can be debated properly and new experimentation undertaken.

We have added a note to the end of our discussion to highlight that these other works are in review and will likely appear shortly.

Reviewer 3

During translation initiation, the GTP-bound form of eukaryotic initiation factor 2 (eIF2) delivers the initiator methionyl-tRNA (Met-tRNA^{iMet}) to the 40S ribosomal subunit and then dissociates as a GDP-bound form. GDP-eIF2 is then recycled into GTP-eIF2 by the guanine nucleotide exchange factor (GEF) eIF2B for assembly of a new ternary complex (TC) with Met-tRNA^{iMet} for the next round of translation initiation.

Under various stressful conditions, cells reprogram their translational machineries, partly by activating an integrated stress response (ISR) that leads to phosphorylation of the α -subunit of eIF2 on the residue Ser51. Phosphorylated eIF2 (eIF2 α P) has been known to bind to eIF2B very tightly, thus reducing its nucleotide exchange activity and consequently reducing the reassembly of the TC. The reduced level of TC decreases the average rate of translation; however, increases the rate of translation of a specific subset of mRNAs in order to adapt to stress. A large amount of genetic and biochemical data support the functional interaction between eIF2 and eIF2B; however, the structural data have been lacking.

Both eIF2 and eIF2B are multi-subunit proteins. eIF2 consists of three α , β and γ , whereas eIF2B consists of five α , β , γ , δ and ϵ . A few structures of eIF2 (e.g., PDB ID = 3JAP) and eIF2B (e.g., PDB IDs = 5B04, 6CAJ and 6EZO) are known. The crystal structure of eIF2B of the fission yeast *S. pombe* reveals that two copies of each of these subunits assemble into a decamer (PDB ID = 5B04). Like the fission yeast eIF2B, the cryo-EM structure of human eIF2B has also shown to adopt a decamer in the presence of a small molecule ISRIB (PDB ID = 6CAJ and 6EZO). In this manuscript, authors have resolved two cryo-EM structures of eIF2B purified from the budding yeast *S. cerevisiae*: one bound to phosphorylated eIF2 (**Fig 1**) and other to non-phosphorylated eIF2 (**Fig 5**). They have observed that the subunit structures of the budding yeast eIF2B in the complex of eIF2B/eIF2 or eIF2B/eIF2 α -P are very similar to the published decameric structure of the fission yeast eIF2B or human eIF2B. However, their structures provide direct evidence that the eIF2 α subunit contacts mainly with the regulatory α and δ subunits of eIF2B (**Supplemental Fig 4**).

Moreover, the complex of eIF2B/eIF2 or eIF2B/eIF2 α -P reveals that the Ser51 phosphorylation site and the KGYID motif of eIF2 α make the core contacts with the eIF2B α subunit (**Fig 3 and Supplemental Fig 5**). These observations are consistent with the prior molecular genetics and biochemical studies that suggest that the residues surrounding the Ser51 phosphorylation site and the KGYID motif of the eIF2 α interact with the eIF2B complex. To provide further genetic evidence for their structure, authors analyze the interacting surfaces of both eIF2 α and eIF2B α , mutate a few surface residues, and show that a single mutation in eIF2 α (e.g., I63N) or eIF2B α (e.g., T41A) exhibited a GCN⁻ phenotype (**Supplementary Fig 5**). Finally, authors propose a model on how eIF2B catalyzes the exchange of GTP for GDP. They propose that binding of eIF2 α to eIF2B triggers a conformation change in the D3 domain of eIF2 α bound to eIF2B $\beta\gamma$ complex, moving away from the catalytic eIF2B $\gamma\epsilon$ subunits (**Fig 2 and Fig 6**). These studies significantly advance our knowledge on translational control by eIF2 α phosphorylation.

We thank the reviewer for their thoughtful and helpful review of our manuscript.

Some minor comments:

1. This reviewer had to read a couple of times to understand the **Fig 2**. The legend of **Fig 2** should be described clearly. How did they align two structures? What is the reference? Difficult to understand the lines - "aligned onto eIF2 α domain 3 TC from 3JAP"... b-g "aligned modelled onto 3JAP eIF2 (as in panel a)"...

We have reworded the legend to make the process clearer. This analysis used standard tools in UCSF-Chimera.

new legend wording:

Fig 2a: eIF2 α conformation differs between eIF2 α P/eIF2B and TC complexes. eIF2 α from our eIF2 α P/eIF2B (domains 1 and 2 shown in gold and arrowed, domain 3 in grey) aligned onto TC from 3JAP (2 α domains 1-3 in grey) using eIF2 α domain 3 as a reference.

2. Authors suggested an elbow-like rotation of eIF2 α domains 2 and 3 when eIF2B binds to eIF2 (**Fig 2** and **Fig 6**). This reviewer was wondering two things.

(I) Did authors try to resolve the structure of eIF2B without its α -subunit?

No we have not attempted to solve this structure, but it would be interesting to do in the future.

(II) How can it be explained the observation that the eIF2B γ arm is relatively far apart from the eIF2 α domain 3 in the complex of eIF2B/eIF2?

The density of eIF2B γ is the weakest of all the subunits (see the Local resolution plot in Figure 6C). We are only able to resolve density for the amino terminal domain and not the Left-handed β helical domain. The N-terminal domain is positioned close to eIF2 γ . The eIF2B γ C-terminal domain must be present as it is where the Flag-tag is located that is used to purify the eIF2B complex. This domain was not resolved in prior EM structures of human eIF2B bound to ISRIB, suggesting it is mobile.

Additionally, the eIF2 α domain 3-eIF2 γ density is rigid body fitted in the consensus position. However the eIF2 α inter-domain flexibility we describe for eIF2 α P/eIF2B also applies to the eIF2/eIF2B structure, so while we show one average position in the final model, there are conformations that place eIF2 α domain 3 and eIF2B γ closer to each other.

3. Line 227: eIF2B α -T41 and eIF2B α -E44 approach eIF2 α -Y82. What do authors mean by stating the word “approach”? What’s the distance between residues eIF2B α -T41 and eIF2 α -Y82 or residues eIF2B α -E44 and eIF2 α -Y82.

We cannot resolve hydrogen atoms and water molecules at this resolution. Measuring C-C and C-O distances shows these are around 4 Å.

The measured atom center distances in Chimera are:

2B α -2 α

E44-Y82 is 3.83 Å

T41-D84 is 3.50 Å

2Bdelta-2 α

E377-I59 = 3.76 Å

L381-I63 = 4.21 Å

In the manuscript we have edited the sentence containing the word ‘approach’. it now states:

However, in eIF2B α side chains of T41 and E44 contact eIF2 α D84 and Y82, respectively (non-H atoms are within 4 Å). Both eIF2 α residues are within the important KGYID element.

4. The phosphorylated side chain of the residue eIF2 α -Ser51 appears to be inside the structure (**Supplemental Fig 5C**). If that is correct, how will you connect the physiological relevance of Ser51 phosphorylation?

The Ser52 residue is surface exposed within the complex. Previous structural studies indicate that the loop containing the phosphorylated residue is flexible. Here the added negative charge of phosphorylation appears to stabilize the loop conformation as we describe on page 8:

The positively charged side chains R54 and R64 are angled towards Ser52(P) and likely help stabilise this conformation of this important loop of eIF2 α . All eIF2 α residues in contact with the eIF2B core are conserved between the yeast and human proteins.

In response to the comment we have reworded discussion sentences on p15/16:

While the precise mechanism of how eIF2 α P inhibits GEF action is not yet resolved, one possibility consistent with the available data³ is that phosphorylation stabilizes the phospho-loop conformation to permit tighter binding affinity to eIF2B. A failure to release eIF2 therefore locks eIF2(α P)/eIF2B together and hence sequesters eIF2B (grey arrows in Fig. 6). As eIF2B is always found in limiting concentrations in vivo. A lack of free eIF2B limits TC formation impairing general protein synthesis initiation, but activating the ISR.

5. In **Fig 4**, authors show that PKR and eIF2B compete for binding to eIF2 α . Authors also show that Ser51 phosphorylation of eIF2 α -I63N mutant protein by GCN2 was not affected (**Supplemental Fig 5b**), but the eIF2 α -I63N mutant eliminated the GCN2-mediated translational control when eIF2B α was absent or mutated at the residue eIF2B α -T41 or -F73 (3-AT sensitive phenotype, **Supplemental Fig 5a**). Does the eIF2B complex compete with PKR when it carries an eIF2B α -T41A mutation or when the complex is devoid of the eIF2B α ?

Prior genetics and biochemistry suggests that these mutant forms do not compete in the same way. High-level expression of PKR in yeast cells from a Gal promoter is lethal to wild-type yeast, but T41A cells are not affected at all. The eIF2B α deleted cells also grow, but not as well as the T41A cells. See Figure 4B in reference 27. As indicated as suggested in reviewer's point 2 above, future work to determine the structure of mutant forms of eIF2B with eIF2 will likely be of interest, especially given the potential that alternative interactions may be possible between eIF2 and eIF2B—see reviewer 2 comments and response.

6. The discussion on the viral translational control and the PKR kinase inhibition by viral protein K3L in the last section are highly speculative and based on the observation upon superposition of K3L structure on eIF2 α in the complex of eIF2B (**Supplemental Fig 8**). Please comment on how would you interpret observation that the corresponding yeast eIF2B α residues Y304 and D305 are absent in its human ortholog when aligned them.

The reviewer is correct that the extreme C-termini differ between yeast and human eIF2B α . To establish if this difference may impact on the observations we show in Supp. Fig. 8 we have now superimposed the published human eIF2B structure into this model. Here human eIF2B was aligned to our yeast eIF2B using the eIF2B α subunit as a reference, with K3L aligned to eIF2 α as in Supplementary Figure 8.

The Figure below shows the results of this analysis—displaying only human eIF2B and K3L for clarity.

The clashes reported in our manuscript with K3L K45 and M48 are also evident here with the equivalent human eIF2B α residues, showing that binding of K3L to eIF2B at this site should not be possible. We have therefore left this section unchanged.

REVIEWERS' COMMENTS:

Reviewer #1 (Remarks to the Author):

I am fully satisfied by the response of the authors and the revised version of the manuscript; I would therefore strongly support publication of this study.

Felix Weis

Reviewer #2 (Remarks to the Author):

The authors have satisfied my most serious concern about the eIF2:eIF2B complex. If there was no subset of particles belonging to the alternative complex observed by the Walter and Ito groups, then the results are what they are. Therefore, I have no major concerns.

Minor point: please, discuss which cross-links from Kashiwagi et al., 2016, are consistent with the structures reported here, and which are not, in particular as it relates to cross-links between eIF2alpha and eIF2Bbeta, and between eIF2gamma and eIF2Bepsilon.

Reviewer #3 (Remarks to the Author):

The revised version of the manuscript is substantially improved. I believe that this paper addresses an important question in the field of translation and is of high interest to readers of Nature Communications.

REVIEWERS' COMMENTS:

Reviewer #1 (Remarks to the Author):

I am fully satisfied by the response of the authors and the revised version of the manuscript; I would therefore strongly support publication of this study.

Felix Weis

We thank the reviewer for their review

Reviewer #2 (Remarks to the Author):

The authors have satisfied my most serious concern about the eIF2:eIF2B complex. If there was no subset of particles belonging to the alternative complex observed by the Walter and Ito groups, then the results are what they are. Therefore, I have no major concerns.

Minor point: please, discuss which cross-links from Kashiwagi et al., 2016, are consistent with the structures reported here, and which are not, in particular as it relates to cross-links between eIF2alpha and eIF2Bbeta, and between eIF2gamma and eIF2Bepsilon.

We thank the reviewer for their review. We have included extra sentences discussing the cross-links between eIF2 and its partners including eIF2B beta and eIF2B epsilon. As this discussion is detailed and has the potential to distract readers from the main points being made, we have moved it to a new supplementary discussion section within the supplementary material. This new text contains this both a discussion of the cross-linking data and also a slightly expanded version of the 'addendum' text that we appended to the end of the discussion in the previous revised version.

Reviewer #3 (Remarks to the Author):

The revised version of the manuscript is substantially improved. I believe that this paper addresses an important question in the field of translation and is of high interest to readers of Nature Communications.

We thank the reviewer for their review